# Advancing Lithium-Ion Batteries’ Electrochemical Performance: Ultrathin Alumina Coating on Li(Ni_0.8_Co_0.1_Mn_0.1_)O_2_ Cathode Materials

**DOI:** 10.3390/mi15070894

**Published:** 2024-07-09

**Authors:** Mehdi Ahangari, Fan Xia, Benedek Szalai, Meng Zhou, Hongmei Luo

**Affiliations:** Department of Chemical and Materials Engineering, New Mexico State University, Las Cruces, NM 88003, USA; mahani92@nmsu.edu (M.A.); fxia@nmsu.edu (F.X.); szalaib@nmsu.edu (B.S.)

**Keywords:** Ni-rich cathode, surface coating, lithium-ion batteries, atomic layer deposition

## Abstract

Ni-rich Li(Ni_x_Co_y_Mn_z_)O_2_ (x ≥ 0.8)-layered oxide materials are highly promising as cathode materials for high-energy-density lithium-ion batteries in electric and hybrid vehicles. However, their tendency to undergo side reactions with electrolytes and their structural instability during cyclic lithiation/delithiation impairs their electrochemical cycling performance, posing challenges for large-scale applications. This paper explores the application of an Al_2_O_3_ coating using an atomic layer deposition (ALD) system on Ni-enriched Li(Ni_0.8_Co_0.1_Mn_0.1_)O_2_ (NCM811) cathode material. Characterization techniques, including X-ray diffraction, scanning electron microscopy, and transmission electron microscopy, were used to assess the impact of alumina coating on the morphology and crystal structure of NCM811. The results confirmed that an ultrathin Al_2_O_3_ coating was achieved without altering the microstructure and lattice structure of NCM811. The alumina-coated NCM811 exhibited improved cycling stability and capacity retention in the voltage range of 2.8–4.5 V at a 1 C rate. Specifically, the capacity retention of the modified NCM811 was 5%, 9.11%, and 11.28% higher than the pristine material at operating voltages of 4.3, 4.4, and 4.5 V, respectively. This enhanced performance is attributed to reduced electrode–electrolyte interaction, leading to fewer side reactions and improved structural stability. Thus, NCM811@Al_2_O_3_ with this coating process emerges as a highly attractive candidate for high-capacity lithium-ion battery cathode materials.

## 1. Introduction

Rechargeable lithium-ion batteries (LIBs) are pivotal in the evolution of microelectronics and serve as primary power sources for portable electronic devices. Their superior energy density, both in terms of weight and volume compared to other rechargeable battery technologies, has made them ubiquitous, integral, and essential components of modern life [1,2,3]. The burgeoning demand for electric vehicles (EVs) and hybrid electric vehicles (HEVs) has positioned LIBs as a promising solution to meet the requirements for high energy and power density [4,5]. Among all types of cathode materials, LiNi_x_Co_y_Mn_z_O_2_ (NCM, where x + y + z = 1)-layered oxide materials have attracted attention due to their low cost, high capacity, and long stability [6]. In particular, Li(Ni_0.8_Co_0.1_Mn_0.1_)O_2_, as a crucial component of lithium-ion batteries, has garnered considerable attention due to its high reversible specific capacity (200 mAh g^−1^) resulting from the double-redox reaction of Ni^2+^/Ni^4+^ and its relatively favorable cost profile [7]. Following the commercial success of NCM with moderate nickel content, it has been demonstrated that producing Ni-rich NCM-layered oxides (≥0.8) with higher nickel and lower cobalt content is an effective strategy to enhance the cathode’s specific capacity and operating voltage while reducing costs [8,9]. This approach is particularly well-suited for practical applications in the EV sector. Nevertheless, the practical adoption of Ni-enriched NCM materials faces challenges such as rapid capacity decay and increased impedance upon cycling due to pronounced structural instability and an unstable electrode–electrolyte interface [10,11,12]. The main mechanisms for electrochemical degradation are as follows: (i) irreversible migration of Ni ions to Li sites due to the similar ionic radii of Ni^2+^ (0.69 Å) and Li^+^ (0.76 Å), leading to Ni/Li mixing [13]; (ii) the reaction of residual lithium on the surface with moisture in the air, forming passivating layers such as Li_2_CO_3_ and LiOH on the cathode surface [14]; (iii) the dissolution of transition metals, resulting in interfacial resistance and capacity deterioration; and (iv) irreversible detrimental phase transitions among three hexagonal structures (H1, H2, and H3) that manifest when Ni-rich cathode materials undergo repeated Li insertion/extraction, particularly during highly charged/discharged states, leading to microcrack propagation and pulverization [15,16]. The mentioned drawbacks hinder the commercialization of high-Ni-content cathode materials in LIBs.

Surface modification has emerged as a great strategy for enhancing the electrochemical performance of Ni-rich cathode materials and resolving the mentioned technical issues, as a continuous coating can effectively decrease the occurrence of parasitic surface reactions and stabilize both surface and bulk structure of the materials [17,18]. A suitable coating is characterized by properties such as conformality, low thickness, integrity, and continuity [19]. Metal oxides such as TiO_2_ [20], ZrO_2_ [21], Al_2_O_3_ [22], and MgO [23] are the most common materials used for their low cost and high protection properties. Among these, Al_2_O_3_ has been widely utilized as a coating for cathode active materials in LIBs, which can be applied using wet chemistry or atomic layer deposition (ALD) techniques [24]. The ALD method has earned a good reputation for producing ultrathin films with atomic-level control using sequential, self-limiting surface reactions [25]. It has been reported that ZrO_2_ [26], TiO_2_ [27], and MgF_2_ [28] coatings applied to cathode active materials using ALD have enhanced the stability of the surface-modified samples.

Herein, an ultrathin Al_2_O_3_ coating (2 nm) using an ALD machine was adopted for the surface of NCM811 cathode active material for LIBs using trimethylaluminum (TMA) and H_2_O in a process consisting of two half-reaction steps. The resultant Al_2_O_3_-coated NCM811 was assembled as cathode electrodes, demonstrating significantly improved stability compared to pristine electrodes upon prolonged cycling.

## 2. Materials and Methods

### 2.1. Materials Synthesis

NCM811 powder was produced by the solid-state method. All the reagents used in this research were purchased from Sigma-Aldrich company (St. Louis, MO, USA). A mixture of high-purity NiSO_4_.6H_2_O, CoSO_4_·7H_2_O, and MnSO_4_·H_2_O was dissolved in distilled water (1 mol/L), maintaining a molar ratio of Ni:Co:Mn = 8:1:1. Then, a 2 mol/L NaOH solution was added to the transition metals solution under a N_2_ atmosphere while mixing the two solutions. The pH was adjusted to 10.0–11.0 using NH_3_·H_2_O, and the temperature was maintained at 50 °C for 24 h. The resultant powder was then filtered and washed using distilled water. Following this, the obtained powder (Ni_0.8_Co_0.1_Mn_0.1_(OH)_2_) was kept in a vacuum oven at 120 °C for 24 h. Finally, Ni_0.8_Co_0.1_Mn_0.1_(OH)_2_ and LiOH.H_2_O were mixed at a 1.00:1.05 molar ratio and ball-milled, followed by calcination of the mixture at 750 °C for 12 h under a pure oxygen atmosphere in a tube furnace.

Al_2_O_3_ on NCM811 was performed in an Angstrom-Dep ALD machine (Albuquerque, NM, USA), which was equipped with a rotary reactor. NCM811 powder served as the active material to be coated with Al_2_O_3_. Nine grams of NCM811 powder were loaded into the rotary reactor. N_2_ gas was employed as the carrier and purging gas, with the temperature of TMA and water set at 90 °C, while the batch reactor was maintained at 120 °C. The two half-reaction steps consisted of injecting TMA and H_2_O with 0.2 and 0.5 s purging, respectively, between each pulse, and purging of 30 s after the last pulse. Three ALD growth cycles were used to prepare the Al_2_O_3_-coated powders. However, the resulting film thickness was 2 nm, which is approximately 6–7 times thicker than expected from ALD. This suggests that chemical vapor deposition (CVD) might contribute significantly to the growth process. The deviation in thickness is likely due to the short pulse and purge times used during the process.

### 2.2. Material Characterizations

Powder X-ray diffraction (XRD) analysis was conducted using a Bruker AXS D8 (Billerica, MA, USA) Focus diffractometer equipped with a LynxEye position-sensitive detector (PSD), utilizing Cu Kα radiation (λ = 0.15406 nm) and a 0.2 mm slit (Empyrean XRD PANalytical, UK) on pristine and alumina-coated NCM811. The diffraction pattern was recorded in the 2θ range of 10–80° with a step size of 0.02° and a count time of 3 s per step. Morphological characteristics and particle size were examined using a field emission scanning electron microscope (Hitachi FESEM Model SU7000, Tokyo, Japan) equipped with an elemental energy dispersive spectroscopy (EDS) detector. The chemical composition of samples was determined by EDS. A transmission electron microscope (TEM) operated at 200 kV was employed to identify the thickness and continuity of the coating on the powder’s surface.

### 2.3. Electrochemical Measurements

The as-prepared materials underwent electrochemical measurements in CR2032 coin cells at room temperature. Cathode electrodes were prepared using a slurry coating procedure. The slurry comprised 80 wt.% NCM811 (active material), 10 wt.% Super-P carbon black (conductive agent), and 10 wt.% polyvinylidene difluoride (PVDF, binder) dissolved in N-methyl-2-pyrrolidene (NMP). The solution was stirred overnight with a magnetic stirrer, and the cathode electrodes were fabricated by tape-casting the mixed slurry onto battery-grade aluminum foil using the doctor blade method. After tape casting, the cathodes were dried overnight at 120 °C in a vacuum oven. The CR2032 coin cells were assembled in an argon-filled glove box under a dry argon atmosphere and <0.1 ppm water and oxygen. The electrolyte used was 1 M LiPF_6_ dissolved in a 1:1 mixture of ethylene carbonate (EC) and diethyl carbonate (DEC). Lithium foil served as the anode, and a Celgard 2400 membrane acted as the separator. The cathode electrode mass loading was approximately 2.3–2.5 mg cm^−2^. Following assembly, the cells rested for 10 h before electrochemical characterization.

For the rate capability test, cells were charged in the galvanostatic mode to cutoff voltages of 4.3, 4.4, and 4.5 V with varying current densities (0.1, 0.2, 0.5, 1.0, 2.0, and 5.0 C, 1 C = 200 mAh g^−1^, where g refers to the mass loading of the active material) followed by discharging to 2.8 V at the same rate as charging using a NEWARE battery testing machine. A 10 min resting period was applied prior to each step. Long-term cycling was conducted at 1 C for 150 cycles. The half-cells were cycled at 0.1 C for 3 cycles before cycling at 1 C.

Three-electrode cells were employed for cyclic voltammetry (CV) and electrochemical impedance spectroscopy (EIS), with lithium foil as the reference and counter electrode in a three-electrode cell system. Cyclic voltammograms of the electrodes were recorded within the potential range of 2.8–4.6 V at a scanning rate of 0.1 mV s^−1^. The samples were cycled at 0.05 C for 3 cycles before CV measurements. EIS measurements were performed in a frequency range of 100 kHz to 10 mHz with a perturbation amplitude of ±10 mV when the samples were charged to the upper cutoff voltage. The measurements were conducted using a CHI 660E electrochemical workstation.

## 3. Results and Discussion

### 3.1. X-ray Diffraction Analysis

Pristine and coated samples were analyzed by XRD to evaluate crystal ordering and determine any impurities and phase changes during powder synthesis and the Al_2_O_3_ deposition process. As shown in Figure 1, the XRD patterns of both samples indicate that the powder structures are well indexed to the layered hexagonal α-NaFeO_2_ type belonging to the R-3m space group. There are no significant shifts in peak positions or any new phases related to impurities. The I(003)/I(104) ratio of both samples is approximately the same and exceeds 1.2, 1.78, and 1.71 for uncoated and coated samples, respectively. Any change in the I(003)/I(104) ratio is attributed to changes in lithium content in the unit cell, which typically occurs during additional heat treatment in the coating process [29]. It is generally accepted that a peak intensity ratio greater than 1.2 signifies a good layered structure and a lower degree of Li^+^/Ni^2+^ cation mixing in the lattice [30]. Moreover, the obvious splitting of the (006), (102) and (108), (110) peaks indicates the maintenance of a well-ordered layered structure as well as a high degree of crystallinity [31]. Additionally, there is no shift in the (003) reflection in the coated sample, which significantly implies two points: first, powder synthesis is highly accurate, and second, the Al_2_O_3_ coating process has no effect on the host crystal structure of NCM811 powder and does not introduce a new phase into the particles. This might be related to the fact that the coating deposition process was performed at a low temperature (120 °C), which inhibited Al diffusion into the lattice structure, ensuring no changes occurred in the crystal structure during the coating synthesis. The distinct diffraction peak of the Al_2_O_3_ phase is not detected in the NCM811@Al_2_O_3_ XRD pattern. This suggests that an ultrathin alumina coating covered the NCM811 particle surfaces and that the coating has a very low content and is an amorphous structure, which would be expected for a growth temperature of 120 °C.

### 3.2. Morphology

SEM images (Figure 2a,b) were captured to analyze the surface morphology and microstructure of the samples. Both samples exhibited secondary microspherical-like shape particles due to the aggregation of numerous nanometer-sized primary particles. Notably, the surface morphology of NCM811 particles remained unchanged, indicating a thin Al_2_O_3_ layer. EDS mapping of transition metals and Al (Figure 2c,d) was conducted to determine the element distribution on the particle surface. The images confirm the homogeneous distribution of transition metals and Al in the coated sample, indicating the desirable conformality and uniformity of the applied coating. TEM observation (Figure 2e) reveals a smooth and uniform Al_2_O_3_ coating achieved on the NCM811 particles, with an ultrathin thickness. The coating thickness is critical; too thick a coating inhibits Li^+^ ion intercalation and extraction, while too thin a coating is ineffective in protecting the cathode material from reactions with the electrolyte [32]. More measurements for the coating thickness are provided in Appendix A.

### 3.3. Electrochemical Properties

The impact of Al_2_O_3_ using the ALD coating system on the enhancement of the electrochemical performance of the NCM811 cathode material was assessed through cycling at 1 C across the voltage ranges of 2.8–4.3, 4.4, and 4.5 V vs. Li/Li^+^ at 25 °C, as depicted in Figure 3. The results reveal that the coated sample demonstrated superior electrochemical performance across all upper cutoff voltages. Particularly noteworthy is the significant difference in capacity retention as the upper cutoff voltage increased, with values of 82.00%, 75.68%, and 63.64% for pristine NCM811 compared to 87.00%, 84.79%, and 74.92% for the modified sample at 4.3, 4.4, and 4.5 V, respectively. Furthermore, both samples exhibited Coulombic Efficiency (CE) exceeding 95%. At the critical 4.5 V cutoff voltage, the Al_2_O_3_-coated sample delivered a discharge capacity of 198.93 mAh g^−1^ at 1 C discharge current, slightly lower than pristine NCM811, which yielded 200.66 mAh g^−1^. After 150 cycles, the former delivered 149.04 mAh g^−1^, while the latter achieved 127.69 mAh g^−1^. The diminished capacity retention of the uncoated sample compared to the coated counterpart can be attributed to the high polarization and deterioration of the interface structure of NCM811, particularly under high upper cutoff voltage operation [33]. Indeed, the uniform nano-Al_2_O_3_ coating reduced the electrode–electrolyte reaction at the interface and delayed the formation of a thick passive layer, thereby postponing polarization at the material interface. Additionally, the formation of HF in the electrolyte leads to damage to cathode materials, while an ultrathin layer can significantly prolong the onset of severe degradation of the active material [34]. The relevant electrochemical stability results corresponding to this experiment are provided in Table 1. The data in Table 1 suggest that although the discharge capacity of the samples in the first few cycles is higher at 4.5 V cutoff potential, the capacity loss is comparable. Therefore, an operating voltage of 4.4 V can be identified as the optimized working condition for NCM811@Al_2_O_3_, striking a balance between capacity retention and high-capacity delivery. One plausible explanation for the poor cycling stability of NCM811 when charging to a 4.5 V potential is electrolyte oxidation and the side reactions between the cathode and electrolyte [35].

Voltage fading in Ni-rich NCM cathode materials occurs during cycling, primarily due to the destruction of the crystal structure and irreversible phase transitions. Galvanostatic voltage profiles recorded with a discharge current density of 1 C for both pristine and coated samples for the 1st and 150th cycles are illustrated in Figure 4 to compare the voltage decay over cycling. It is evident that, for all samples, the discharge profiles gradually shifted to lower voltage plateaus during cycling, accompanied by a decrease in discharge capacity. This phenomenon is attributed to the dissolution of transition metals from the active material due to structural and interfacial instabilities [36]. Moreover, the voltage fading increased as the upper cutoff voltage increased. Specifically, the pristine NCM811 experienced voltage fading of 0.0986 V, 0.1559 V, and 0.3522 V during cycling at cutoff voltages of 4.3 V, 4.4 V, and 4.5 V, respectively. In contrast, the Al_2_O_3_-coated sample exhibited lower voltage decay in this experiment, with values of 0.0710 V, 0.1413 V, and 0.2930 V for the corresponding cutoff voltages. The reduced voltage decay during cycling can be attributed to the successful suppression of polarization in the NCM811 electrode by the Al_2_O_3_ coating, which improved the structural stability and phase reversibility of the active material while protecting the bulk from direct contact with the electrolyte [37].

The results of rate capability performance for the samples at various charging cutoff voltages are shown in Figure 4. The samples were charged and discharged at current densities ranging from 0.1 to 5 C. Due to the low diffusion efficiency of Li^+^ at high current densities, the discharge capacity of both cathode materials decreased as the discharge current density increased. However, the coated sample exhibited higher discharge capacity, especially at higher rates (2 and 5 C) and upper cutoff voltages (4.4 and 4.5 V).

At low current densities and in the initial cycles (1–25), both samples demonstrated similar reversible discharge capacity. In contrast, at higher rates (5 C) and in later cycles (26–55), the coated sample delivered more capacity, attributed to the improved stability of the surface structure. For instance, at the 4.5 V charging cutoff, pristine and coated NCM811 exhibited a reversible capacity of 217.28 and 213.54 mAh g^−1^ in the first cycle, respectively. The difference in initial discharge capacity might be due to the electrochemical inactivity of Al_2_O_3_. Over multiple charging and discharging cycles at different rates, the Al_2_O_3_-coated NCM811 demonstrated an impressive reversible capacity of 210.90 mAh g^−1^, compared to 202.16 mAh g^−1^ for the pristine NCM811.

It is worth noting that while the Al_2_O_3_ coating did not significantly inhibit Li^+^ ion diffusion at different rates, it enhanced the stability of the sample, enabling it to deliver more capacity than the pristine sample, irrespective of the charging voltage. The cycling and rate performances of the samples confirm that the structural stability of NCM811 cathode materials has been effectively improved by the Al_2_O_3_ coating.

### 3.4. EIS Measurements

An EIS test was conducted to further investigate the positive influence of the Al_2_O_3_ coating on the NCM811 cathode material. The Nyquist plots of the samples and the equivalent electrical circuit at the charged state at various upper cutoff voltages before cycling and after 150 cycles are illustrated in Figure 5. Each EIS plot consists of two semicircles and a straight line in different frequency regions. In the circuit, R_s_, R_SEI_, and R_ct_ represent the electrolyte resistance, the film resistance due to the solid electrolyte interface, and the charge transfer resistance related to the interface between the electrolyte and the electrode, respectively. In addition, the straight line at a low frequency corresponds to Warburg impedance (W), which is related to Li^+^ diffusion in the particle [38].

The R_s_ value for all samples is nearly the same, as identical electrolytes were used for this experiment. The R_ct_ value is listed in Table 2. Although the R_ct_ values for both electrodes were nearly the same before cycling, the value increased after cycling. However, the increase in charge transfer resistance was significantly higher for the uncoated sample compared to the coated one. This sharp increase in R_ct_ correlates with the surface distortion induced by side reactions with the electrolyte. In contrast, the alumina-coated NCM experienced a lower increase in R_ct_ indicating a superior charge-transfer rate during long-term cycling. The EIS results are consistent with the cycling and rate capability results. It is worth noting that cycling at a 4.5 V upper cutoff potential induced a significant increase in R_ct_, which can justify the drastic capacity retention deterioration over cycling.

### 3.5. Cyclic Voltammetry Measurements

To further understand the electrochemical behavior of NCM811 and NCM811@Al_2_O_3_, cyclic voltammograms of the cathode electrodes were recorded over 15 cycles within a 2.8–4.6 V voltage range at a scan rate of 0.1 mV s^−1^, as shown in Figure 6. The curves for both samples exhibited similar profiles, indicating that the Al_2_O_3_ coating did not participate in the electrochemical reactions. The main paired peaks correspond to the typical phase transitions for layered oxide cathode materials.

The potential difference between the oxidation and reduction peaks is a key kinetic factor influenced by the formation of the cathode solid electrolyte interface or by side reactions on the electrode surface. This potential difference reflects the reversibility of the electrochemical reaction [30]. For the coated sample, the oxidation and reduction peaks are located at 3.848 V and 3.682 V, respectively, with a corresponding potential difference of 0.166 V for the first cycle. In contrast, the uncoated sample has a higher potential difference of 0.191 V.

After 15 cycles, the coated sample exhibited a smaller voltage gap, indicating higher structural stability and a greater degree of reversibility. These results reveal that the ultrathin Al_2_O_3_ coating effectively reduced electrochemical polarization and enhanced electrochemical performance. This improvement is attributed to the coating’s ability to suppress side reactions at the electrode/electrolyte interface.

## 4. Conclusions

In this manuscript, NCM811 cathode active material was synthesized, and an ultrathin Al_2_O_3_ coating was established on the NCM811 surface using the ALD technique. The alumina coating was demonstrated to exist on the surface of the cathode particles without altering their morphology or crystal structure, attributed to the low-temperature coating process. This coating significantly improved the cycling stability and rate capability of the cathode electrode, which is related to the enhanced structural stability of NCM811 powder. Additionally, the coating suppressed irreversible phase transitions and side reactions at the electrode–electrolyte interface, confirmed by EIS and CV measurements. Electrochemical measurements showed that cycling at higher upper cutoff voltages drastically reduced capacity retention for pristine NCM811. However, Al_2_O_3_ surface treatment stabilized the cathode material’s structure, with a 2.8–4.4 V cutoff potential identified as the optimized cycling condition for NCM811@Al_2_O_3_ in this research.

## Figures and Tables

**Figure 1 micromachines-15-00894-f001:**
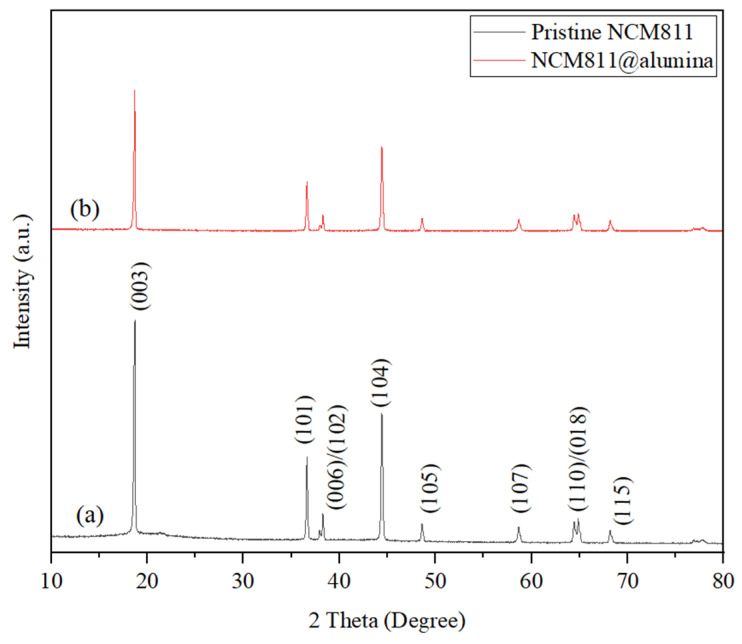
XRD patterns comparing the (a) pristine and (b) NCM811 surface modified with Al_2_O_3_.

**Figure 2 micromachines-15-00894-f002:**
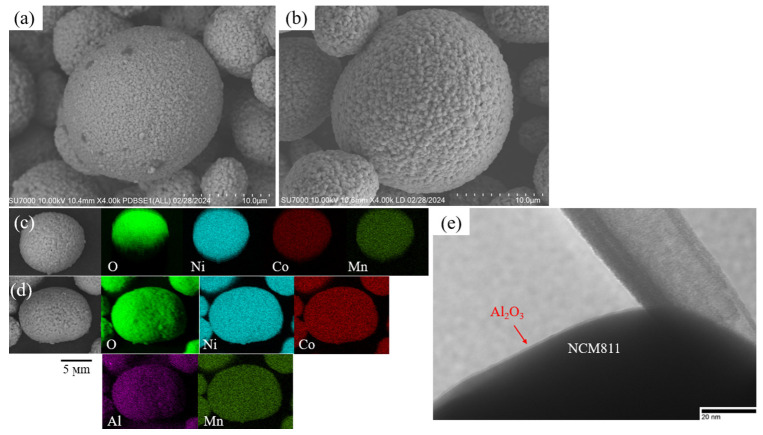
SEM images showing (**a**) pristine and (**b**) NCM811 coated with Al_2_O_3_, along with EDS-mapping of (**c**) pristine NCM811 and (**d**) Al_2_O_3_-coated NCM811 particles. Additionally, TEM images depicting the Al_2_O_3_-coated NCM811 particle are shown in (**e**).

**Figure 3 micromachines-15-00894-f003:**
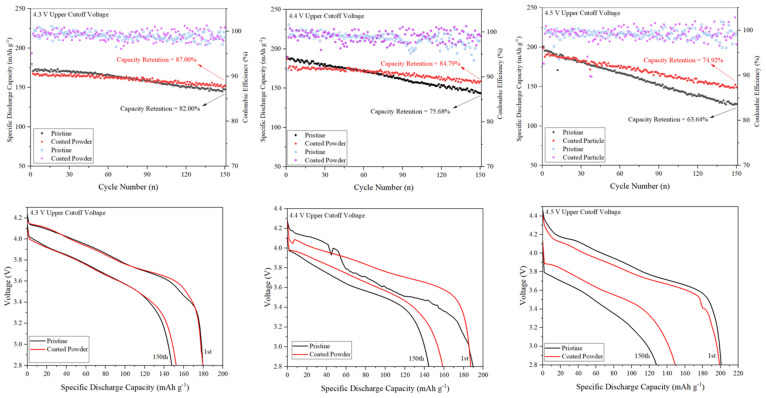
Electrochemical properties of bare and Al_2_O_3_-coated NCM811 over 150 cycles (1 C) at 4.3, 4.4, and 4.5 upper cutoff voltages. The upper figures compare the cycling stability of the samples, while the lower ones illustrate the voltage fading after 150 cycles.

**Figure 4 micromachines-15-00894-f004:**
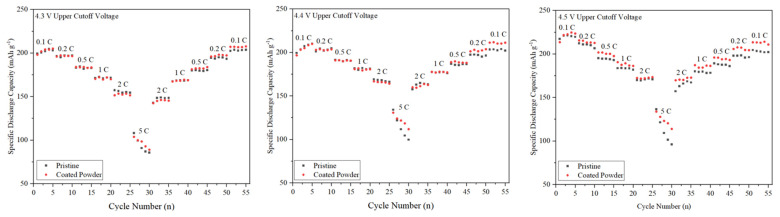
Rate capability for pristine and NCM811@Al_2_O_3_ at different cutoff voltages.

**Figure 5 micromachines-15-00894-f005:**
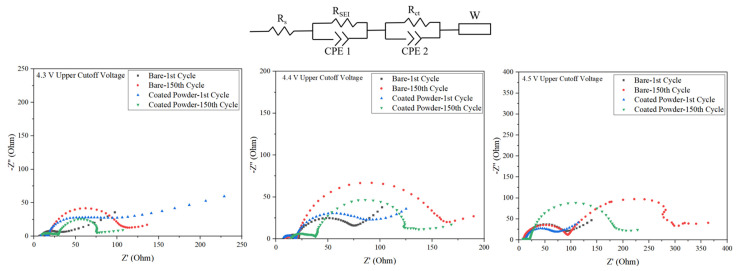
EIS curves of charged bare and Al_2_O_3_-coated NCM811 at 4.3–4.5 voltage range before cycling and after 150 cycles.

**Figure 6 micromachines-15-00894-f006:**
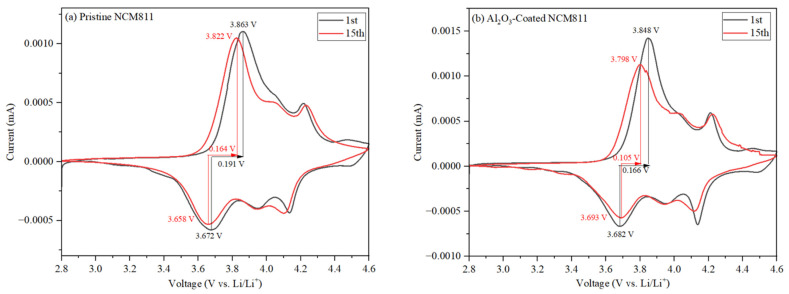
CV curves of (**a**) pristine and (**b**) Al_2_O_3_-coated NCM811 at a scan rate of 0.1 mV s^−1^ within a 2.8–4.6 V voltage range.

**Table 1 micromachines-15-00894-t001:** Electrochemical data for stability of bare and Al_2_O_3_-coated NCM811 over 150 cycles between 2.8 and 4.5 V.

Sample	1st Discharge Capacity (mAh g^−1^)	150th Discharge Capacity (mAh g^−1^)	Capacity Retention (%)	Voltage Fading (V)	Upper Cutoff Voltage (V)
Pristine NCM811	179.46	147.51	82.00	0.0986	4.3
	189.62	144.52	75.68	0.1559	4.4
	200.66	127.69	63.64	0.3522	4.5
Al_2_O_3_-Coated NCM811	179.61	152.25	87.00	0.0710	4.3
	187.23	158.74	84.79	0.1413	4.4
	198.93	149.04	74.92	0.2930	4.5

**Table 2 micromachines-15-00894-t002:** R_ct_ values for pristine and Al_2_O_3_-coated NCM811 before cycling and after 150 cycles.

Sample	R_ct_ before Cycling (Ω)	R_ct_ after 150 Cycles (Ω)	Upper Cutoff Voltage (V)
Pristine NCM811	34.18	114.8	4.3
	75.88	164.00	4.4
	91.34	307.10	4.5
Al_2_O_3_-Coated NCM811	62.59	75.58	4.3
	92.97	126.53	4.4
	72.42	205.8	4.5

## Data Availability

The original contributions presented in the study are included in the article, further inquiries can be directed to the corresponding author.

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
