# Peer review of "Advancing Lithium-Ion Batteries’ Electrochemical Performance: Ultrathin Alumina Coating on Li(Ni0.8Co0.1Mn0.1)O2 Cathode Materials"

_micromachines, 2024, doi:10.3390/mi15070894_

Round 1
Reviewer 1 Report
Comments and Suggestions for Authors
Please see attached document

Minor typos and some spelling errors. Section 2.1 in particular needs revision.
Author Response
Reviewer’s comment 1: This is an interesting paper but there is one major flaw which undermines the whole premise. The three cycle ALD process used here should deliver approximately 0.1 nm per cycle. If the film thickness obtained is 2 nm, this means that the process was not ALD but CVD since otherwise there is no way 2 nm could be obtained for a TMA/water process at 120C. The cycle length is also very short (0.2 s with 0.5 s purging) for a powder process where longer times are typically necessary to enable precursors to reach all surfaces of the powder particles and byproducts to totally evacuate the system to ensure there is no CVD taking place which will affect both thickness and uniformity. This may be the reason why the film is thicker than expected. You need to provide evidence that this is actually an ALD process. For example, using longer pulse and purge times, does the thickness decrease?
Author’s Response: Thanks a lot for the consideration. Generally, coating thickness for powders in ALD highly depends on the various parameters. Mass loading, purging time, and growth cycles are the main factors. The main possible reason for the thickness of the coating in the experiment is the used mass loading (9 g). In the investigated ALD coatings, different mass loadings were used (10-25 g). Therefore, we cannot certainly express that a specific coating would be achieved by the same cycling. For more information, additional information about thickness is available in supplementary information (Figure S1.) to give certainty about the ALD process.
Reviewer’s comment 2: In the XRD measurements, it is claimed that there is little difference between the I(003)/I(104) ratio of coated and uncoated samples [line 147]. However, according to figure 1, this ratio changes from approximately 2.3 for the uncoated to 1.2 for the coated samples. It is also suggested that any changes are due to the coating process by comparison with ref [29]. In this reference a coating temperature of 700 C for 6 h was used whereas the ALD process took only a few minutes at 120 C so it is of no relevance here. This discrepancy has to be properly explained.
Author’s Response: Thanks a lot for the strong comment. In order to increase the accuracy of the results, the research group did the XRD measurements again and in addition, normalization was implemented for, and the background was subtracted. The mentioned problem for the XRD curves has been solved.
Reviewer’s comment 3: In the description of the ALD process, I assume that the purging gas was also N2? This should be explicitly stated.
Author’s Response: Thanks for the comment. The purging gas is mentioned in the manuscript as highlighted:
Line 93: N2 gas was employed as the carrier and purging gas
Reviewer’s comment 4: Line 127 The definition of C is given as C = 200 mAh/g. Specify what the mass, g, refers to.
Author’s Response: This is a common definition for researchers working in the battery area. The “g” in the formula is defined as the reviewer asked and highlighted in the manuscript.
Line 127: 1 C = 200 mAh g−1, where g refers to the mass loading of the active material
Reviewer’s comment 5: Line 161 No XRD peak is seen from the Al2O3. This suggests that it is amorphous which would be expected for a growth temperature of 120 C.
Author’s Response: It is an interesting point that the reviewer mentioned. The additional explanation has been added and highlighted in the manuscript.
Line 164-165: and it is in amorphous structure which would be expected for a growth temperature of 120 °C.
Reviewer’s comment 6: Fig. 2e The dotted lines suggest a film thickness of 7 nm from their positioning. I understand these are just for indication but they need to be more accurate, particularly in light of my earlier comments about film thickness.
Author’s Response: A new TEM image is replaced as the respected reviewers asked for.
Reviewer’s comment 7: Line 192 “…the former delivered 127.69 mAh g-1, while the latter achieved 149.04…” The descriptions ‘former ’and ‘latter’ are in the wrong place since the 127 number refers to the pristine material which is mention after the coated material and the 149 refers to the coated which comes before
Author’s Response: Thanks a lot for this comment. The typo has been modified and highlighted in the manuscript.
Line 195: After 150 cycles, the former delivered 149.04 mAh g-1, while the latter achieved 127.69 mAh g-1.
Reviewer’s comment 8: Line 199-200 “…prolong severe degradation of the active material…” should be “…prolong the onset of severe degradation of the active material…”
Author’s Response: It is corrected and highlighted in the manuscript.
Line 203: prolong the onset severe degradation of the active material
Reviewer’s comment 9: There are a number of typographical and spelling errors which should be corrected, particularly in Section 2.1.
Author’s Response: Thanks a lot for this comment. All the manuscript has been revised to avoid any typos and spelling errors.

Reviewer 2 Report
Comments and Suggestions for Authors
In the manuscript, the authors systematically studied the effects of coating (Al2O3) on NMC811 cathode materials: XRD was first applied to probe the structural evolution after coating, then SEM and TEM were utilized to see how the morphology evolved after the coating of Al2O3; electrochemical tests with increasing cutoff voltage from 4.3, 4.4, 4.5 V vs. Li/Li+ were further implemented; EIS and CV tests were finally implemented and revealed slower growth of resistance and higher reversibility over pristine NMC811 powders.
Given the high quality and encouraging results shown in this manuscript, I recommend accepting this manuscript. Meanwhile, minor revisions are needed before acceptance.
1. For the XRD results shown in Figure 1, can you normalize the results to the strongest <003> peak? I mean, if you normalize the intensity of the <003> peak in Figure 1a and 1b to the same value and adjust the intensity of all other peaks, it is better for the comparison of structural evolution after the coating of Al2O3.
2. For the intensity ratios of I<003>/I<104> for pristine and coated NMC811 powders, the authors mentioned that “1.277 and 1.275 for uncoated and coated samples, respectively”, can you double check the calculation process? From the intensity shown in Figure 1a, I think the I<003>/I<104> ratio for pristine NMC811 powder should be greater than 1.277.
3. For the TEM image shown in Figure 2e, do you have a better one with higher resolution? I think the one shown in Figure 2e is blurred and I suggest replacing it with another one with higher resolution.
4. For the EIS results shown in Figure 5, I think the y axis should be -Z’’. Meanwhile, I think it is better to adjust the x/y ratio to be 1 for the EIS results.
5. For the information revealed by CV results (Figure 6 on page 11/13), the authors claimed that “The main paired peaks correspond to the oxidation/reduction of Ni2+/Ni4+”. I think these main peaks are more related to the typical phase transitions for layered oxide cathode materials. I suggest modifying it.

Author Response
Comments of Reviewer #2:
Highly appreciate your consideration and valuable comments on the paper.
Reviewer’s Comment 1: For the XRD results shown in Figure 1, can you normalize the results to the strongest <003> peak? I mean, if you normalize the intensity of the <003> peak in Figure 1a and 1b to the same value and adjust the intensity of all other peaks, it is better for the comparison of structural evolution after the coating of Al2O3.
Author’s Response: Thanks for the comment. The figures have been normalized and replaced.
Reviewer’s Comment 2: For the intensity ratios of I<003>/I<104> for pristine and coated NMC811 powders, the authors mentioned that “1.277 and 1.275 for uncoated and coated samples, respectively”, can you double check the calculation process? From the intensity shown in Figure 1a, I think the I<003>/I<104> ratio for pristine NMC811 powder should be greater than 1.277.
Author’s Response: The value for the ratio has been recalculated and mentioned in the manuscript. It was a typo, and correct values are replaced (Line 148-149).
Reviewer’s Comment 3: For the TEM image shown in Figure 2e, do you have a better one with higher resolution? I think the one shown in Figure 2e is blurred and I suggest replacing it with another one with higher resolution.
Author’s Response: A new TEM image is replaced as the reviewers asked for.
Reviewer’s Comment 4: For the EIS results shown in Figure 5, I think the y axis should be -Z’’. Meanwhile, I think it is better to adjust the x/y ratio to be 1 for the EIS results.
Author’s Response: x/y ratio adjustment has been done.
Reviewer’s Comment 5: For the information revealed by CV results (Figure 6 on page 11/13), the authors claimed that “The main paired peaks correspond to the oxidation/reduction of Ni2+/Ni4+”. I think these main peaks are more related to the typical phase transitions for layered oxide cathode materials. I suggest modifying it.
Author’s Response: Thanks for the comment. It is more explained and corrected as recommended and highlighted in the manuscript.
Round 2
Reviewer 1 Report
Comments and Suggestions for Authors
I thank you for the various corrections you have made and the supplementary material. However, my main criticism still stands - just because the film is grown in an ALD system, that does not make it ALD. The characteristic of a film grown in ALD conditions is that its thickness is independent of precursor dose either by precursor loading or by pulse and purge times. If it is not consistent with this, it is not ALD. A thickness of 2 nm is equivalent to approximately twenty ALD cycles, not three therefore more than one layer is being deposited in one cycle, that it, it is not ALD. You cannot say that the thickness depends on the precursor dose and that it is also ALD – the two situations are mutually exclusive. What you can say is that the film was grown in in an ALD system but it was not ALD growth because the layer is too thick. If you are willing to remove the claim that this is ALD and admit that there is CVD occurring I will be happy to accept the paper for publication, otherwise I am cannot agree to it.
Author Response
Reviewer’s comment 1: I thank you for the various corrections you have made and the supplementary material. However, my main criticism still stands - just because the film is grown in an ALD system, that does not make it ALD. The characteristic of a film grown in ALD conditions is that its thickness is independent of precursor dose either by precursor loading or by pulse and purge times. If it is not consistent with this, it is not ALD. A thickness of 2 nm is equivalent to approximately twenty ALD cycles, not three therefore more than one layer is being deposited in one cycle, that it, it is not ALD. You cannot say that the thickness depends on the precursor dose and that it is also ALD – the two situations are mutually exclusive. What you can say is that the film was grown in in an ALD system but it was not ALD growth because the layer is too thick. If you are willing to remove the claim that this is ALD and admit that there is CVD occurring I will be happy to accept the paper for publication, otherwise I am cannot agree to it.
Author’s Response: As the respected reviewer recommended, we changed Al2O3 ALD coating to Al2O3-coated sample and ultrathin alumina coating in the manuscript. The changes are highlighted in the manuscript.
Round 3
Reviewer 1 Report
Comments and Suggestions for Authors
In line 97 you state ‘Three ALD growth cycles were used to prepare the Al2O3-coated powders.’ It is essential that you make it clear that this is not an ALD process and the films have a large CVD component to their growth in order not to give any false impressions (and for your own integrity as authors and to show that you understand your own process). This could be done by saying, for example, ‘Three ALD growth cycles were used to prepare the Al2O3-coated powders. However, the film thickness was 2 nm which is approximately 6-7 times thicker than would be expected from ALD indicating that chemical vapour deposition (CVD) forms the major part of the growth process. This is likely to be due to the short pulse an purge times used.’
Author Response
Thanks for the comments. the changes have been made in the manuscript.

Round 4
Reviewer 1 Report
Comments and Suggestions for Authors
In lines 100-101 you have said 'This suggests that chemical vapor deposition (CVD) might contribute significantly to the growth process.' This is still not sufficiently clear. You need to state 'This indicates that chemical vapor deposition (CVD) forms the major part of the growth process.' I will not accept and fudging of this point such as 'might contribute' or 'significantly'. This is not an ALD process and you have to accept that and be explicit in the manuscript.
